# Programmable CRISPR-Cas transcriptional activation in bacteria

Hsing-I Ho[1] , Jennifer R Fang[2] , Jacky Cheung[3] & Harris H Wang[1,4,*]

## Abstract

Programmable gene activation enables fine-tuned regulation of endogenous and synthetic gene circuits to control cellular behavior. While CRISPR-Cas-mediated gene activation has been extensively developed for eukaryotic systems, similar strategies have been difficult to implement in bacteria. Here, we present a generalizable platform for screening and selection of functional bacterial CRISPR-Cas transcription activators. Using this platform, we identified a novel CRISPR activator, dCas9-AsiA, that could activate gene expression by more than 200-fold across genomic and plasmid targets with diverse promoters after directed evolution. The evolved dCas9-AsiA can simultaneously mediate activation and repression of bacterial regulons in *E. coli*. We further identified hundreds of promoters with varying basal expression that could be induced by dCas9-AsiA, which provides a rich resource of genetic parts for inducible gene activation. Finally, we show that dCas9-AsiA can be ported to other bacteria of clinical and bioindustrial relevance, thus enabling bacterial CRISPRa in more application areas. This work expands the toolbox for programmable gene regulation in bacteria and provides a useful resource for future engineering of other bacterial CRISPR-based gene regulators.

**Keywords** bacterial gene regulation; CRISPR tools; protein engineering; synthetic transcription activator

**Subject Categories** Biotechnology & Synthetic Biology; Methods & Resources

**Mol Syst Biol. (2020) 16: e9427**

## Introduction

Transcriptional regulation governs almost every cellular process fundamental to life. In response to cellular or external signals, transcription factors (TFs) in the cell interact with specific DNA sequences to mediate gene activation or repression. A potential path for cellular engineering is therefore the rewiring of transcription factors to alter gene regulatory networks (Isalan *et al*, 2008).

Programmable transcriptional activation and repression in principle offer on-demand control of specific biological processes without the need to permanently alter the genome of a cell. As such, significant past efforts have been devoted to developing synthetic transcription activators by fusing DNA-binding proteins with transcription effector domains to recruit the RNA polymerase (RNAP) complex (Dove & Hochschild, 1998; Joung *et al*, 2000). Unfortunately, these past synthetic TFs generally recognize only predefined DNA sequences and are difficult to reprogram to target other sequences, which greatly limit their utility for transcriptional regulation of diverse endogenous and engineered gene regulatory networks.

With the discovery of new DNA-binding proteins such as Zinc-finger TFs (Beerli & Barbas, 2002), transcription activator-like (TAL) effectors (Joung & Sander, 2013), and CRISPR-Cas systems (Mali *et al*, 2013), there are opportunities to develop next-generation synthetic transcription factors with greater activity and programmability. The Cas9 protein, a member of a large class of RNA-guided DNA nucleases, has emerged over the past several years as a promising system for building synthetic TFs (Bikard *et al*, 2013; Qi *et al*, 2013). Cas9 utilizes a short guide RNA (gRNA) and a protospacer adjacent motif (PAM) sequence on the target DNA to bind a defined sequence based on RNA-DNA base pairing and for cleavage of the target DNA sequence (Sternberg *et al*, 2014). Inactivating Cas9 by mutating the catalytic residues in the nuclease domains results in a nuclease-dead Cas9 (dCas9) that functions solely as a DNA-binding protein. Transcriptional effectors such as activation or repression domains can then be linked to different parts of the dCas9 complex (e.g., dCas9 or gRNA) to enable programmable and targeted transcriptional repression (CRISPRi; Qi *et al*, 2013) or activation (CRISPRa; Maeder *et al*, 2013; Perez-Pinera *et al*, 2013). While a variety of CRISPRi systems has been successfully demonstrated in bacteria (Bikard *et al*, 2013) and eukaryotes (Qi *et al*, 2013) and many mammalian CRISPRa approaches exist (Chavez *et al*, 2016), far fewer successful examples of bacterial CRISPRa have been shown.

In bacteria, sigma factors play a pivotal role in transcriptional initiation machinery (Browning & Busby, 2016). Sigma factors interact with the core RNAP enzyme ($\alpha_2\beta\beta'\omega$) complex and bind to specific promoter sequences. Different types of sigma factors compete for the common pool of core enzymes in bacterial cells and recruit them

1 Department of Systems Biology, Columbia University, New York, NY, USA
2 Department of Biological Sciences, Columbia University, New York, NY, USA
3 Department of Computer Science and Biology, Columbia University, New York, NY, USA
4 Department of Pathology and Cell Biology, Columbia University, New York, NY, USA
*Corresponding author. Tel: +1-212-305-1697; E-mail: hw2429@columbia.edu

to corresponding promoters (Browning & Busby, 2004). Transcription factors further function in *trans* on the holoenzyme and regulate gene expression. Transcription activators usually bind with specific components of the RNAP complex and direct the complex to the target promoter region (Browning & Busby, 2016). However, most transcriptional activation domains in bacteria are not well-characterized and have not been demonstrated to mediate transcriptional activation when coupled synthetically with DNA-binding domains. To our knowledge, only three efforts have been described for engineering bacterial transcriptional activation using CRISPR-Cas. In the first study by Bikard *et al* (2013), dCas9 was fused to the RNAP ω subunit, which interacts with the RNA polymerase to mediate gene activation. However, this CRISPRa system could only function in the ω subunit knockout background (Bikard *et al*, 2013). Deletion of *rpoZ* that encodes ω subunit is known to lead to altered basal transcription profile and fitness defects (Chatterji *et al*, 2007; Weiss *et al*, 2017). Another study used bacterial enhancer binding proteins (bEBPs) as the fused activation domain in a similar approach (Liu *et al*, 2019), but the bEBPs-mediated CRISPRa is only compatible with σ54 promoters and the deletion of endogenous bEBPs is required. Both systems require modification of the bacterial genome, which limits the portability to genetically tractable microbes. Another study by Dong *et al* (Dong *et al*, 2018) used an scaffold RNA (scRNA) containing the gRNA and an MS2 domain, which could bind to an MCP-fused transcription factor SoxS to enable dCas9-mediated transcriptional activation. This system exhibited higher activity after further optimization, but has a narrow targetable region within the promoters (Fontana *et al*, 2020). Furthermore, most of these prior studies have only demonstrated CRISPRa in laboratory *E. coli* strains. The application of CRISPRa on different bacteria species has thus been limited (Peng *et al*, 2018; Yu *et al*, 2018; Lu *et al*, 2019).

To overcome these challenges, we devised a high-throughput platform to screen and select for bacterial CRISPR-Cas transcriptional activators (CasTAs). We first screened a number of natural bacterial and phage regulatory effectors and identified a phage protein that induced gene activation when fused to dCas9. We characterized the targeting window of this CasTA and further performed rounds of directed evolution using our screening platform to yield more effective variants, which can mediate both CRISPRi and CRISPRa of genomic and plasmid targets. We then applied this activator system to a metagenomic promoter library mined from diverse bacteria to build a library of CasTA-inducible promoters of varying basal and induced expression levels that are useful as a resource for the synthetic biology research community. Finally, we describe the successful transfer of our CRISPRa system to other bacterial species of clinical and bioindustrial importance, thus expanding utility to more application areas.

## Results

### A screening–selection platform for bacterial CRISPRa development

To expedite the discovery of bacterial CRISPRa components, we developed a screening–selection platform in *Escherichia coli* to identify candidate dCas9-mediated transcription activators. In our

CRISPRa design, an *S. pyogenes* dCas9 (Qi *et al*, 2013) is C-terminally fused with candidate transcription activation domains or proteins via a previously described flexible peptide linker (SAGGGGSGGGGS) (Chen *et al*, 2013). This CasTA then uses a specific gRNA to target to the regulatory region of a reporter gene for transcriptional activation, gene expression, and production of reporter products (Fig 1A). As such, we separated the three essential components of the platform (i.e., dCas9-activator fusion, the guide RNA, and the reporter gene) into 3 compatible plasmids (Fig 1B). The dCas9 activator was regulated by a $P_{tetO}$ induction system with anhydrotetracycline (aTc) on a p15A medium copy plasmid, while the gRNA was expressed constitutively from a strong promoter (BBa_J23119) on a high copy ColE1 plasmid, and the reporter was placed behind a very weak promoter (BBa_J23117) on a low copy SC101 plasmid (Appendix Fig S1). Since different dCas9 activators may have their own respective optimal gRNA binding windows (Bikard *et al*, 2013; Dong *et al*, 2018) and possible biases toward targetable promoter sequences (Liu *et al*, 2019), the screening–selection platform was designed to be highly modular to facilitate combinatorial assessment of system components. As library-scale screening for transcription activators can often be hampered by auto-activators in the population, we further employed a dual screening–selection reporter design by using both fluorescent protein and antibiotic resistance genes to eliminate potential false positive clones. We further engineered the selective reporter to contain multiple separate antibiotic genes with degradation tags (BBa_M0050) to increase the rate of turnover to reach higher stringency and specificity of the selection (see Materials and Methods, Appendix Fig S2).

Using this platform, we first screened a list of transcriptional activator candidates, including phage proteins, transcription factors, and RNAP interacting proteins (Appendix Table S3), paired with different gRNAs (gRNA-H1, gRNA-H2, gRNA-H3) targeted to different spacing distances to transcriptional start site (TSS) of the reporter gene (59, 81, 118 bp, respectively), for potential dCas9 activators. Among the transcription activation modules screened, we found a phage protein, AsiA, that upregulated the reporter gene expression to a level comparable to the previously identified dCas9-ω activator (Bikard *et al*, 2013), although at a different optimal spacing distance (Fig 1C and D). AsiA (Audrey Stevens' inhibitor A) is a 90 amino acid anti-σ70 protein from the T4 bacteriophage that binds to the host σ70 subunit and suppresses endogenous gene expression (Stevens, 1972; Orsini *et al*, 1993). In combination with another T4 phage protein, MotA, the σ70-AsiA-MotA complex specifically binds to T4 phage promoters and activates phage transcription during the T4 viral life cycle (Minakhin & Severinov, 2005).

When directly fused to dCas9 with a peptide linker, AsiA upregulated gene expression of a GFP reporter, with the tunable magnitude of activation via design of the gRNA targeting positions. Transcriptional activation by dCas9-AsiA (dubbed CasTA1.0) is seen across a wide window along the target regulatory region, reaching up to 12-fold at ~190 base pairs (bp) from the TSS (Fig 1E). In contrast, the optimal gRNA targeting positions for other dCas9 activators (e.g., dCas9-ω and dCas9-MS2/MCP-SoxS) are less than 100 bp from the TSS with a more narrow targetable window (Bikard *et al*, 2013; Dong *et al*, 2018). Unlike other dCas9 activators that mediate activation with re-engineered endogenous transcription factors, AsiA is an anti-σ70 protein that has evolved to outcompete host transcriptional

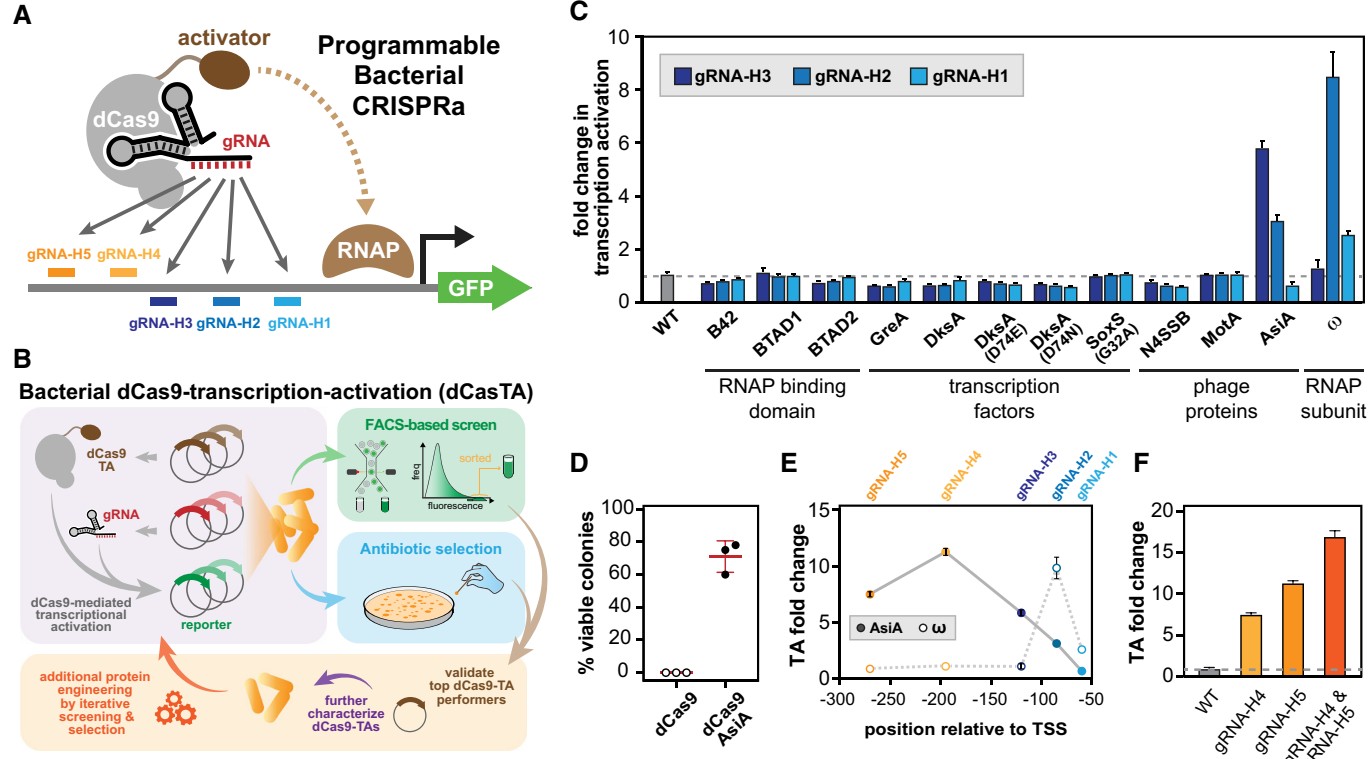

**Figure 1. A high-throughput platform to identify and engineer bacterial CRISPR-Cas transcription activators (CasTAs).**

A  General strategy for bacterial CRISPRa using a dCas9 fused with a transcriptional activator, a targeting gRNA, and reporter genes.

B  System components are constructed in three compatible plasmids. CasTA candidates could be cross validated through GFP and antibiotic resistance gene reporters.

C  Fold induction of the fluorescence of CasTA candidates using different gRNAs targeting to different locations of the GFP reporter gene compared to a strain without CasTA. The dCas9-ω was verified in the ΔrpoZ strain background, and the rest of candidates were verified in the wild-type strain (BW25113).

D  Survival of cells containing upregulated antibiotic resistance reporter induced by dCas9 or dCas9-AsiA with gRNA-H3 under kanamycin selection (2.5 μg/ml).

E  Different gRNAs were paired with dCas9-AsiA and dCas9-ω to profile the optimal gRNA binding distance. The dCas9-AsiA was verified in the wild-type strain (BW25113), and dCas9-ω was examined in the ΔrpoZ strain background.

F  Comparing single and multiple gRNAs with dCas9-AsiA.

Data information: Data in all panels are 3–5 biological replicates with ± standard error of mean (SEM).

machinery. The strong interaction between AsiA and σ70 may result in a different mode of activation from other systems. Simultaneously targeting with multiple gRNAs can further increase transcriptional activation (Fig 1F), although no synergistic enhancement was observed in contrast to eukaryotic CRISPRa systems (Maeder *et al*, 2013).

Based on different CRISPRa architectures that have been described in literature (Chavez *et al*, 2016), we further explored whether AsiA can be tethered to other parts of the dCas9 complex and reach higher activity. The MS2 hairpin RNA has been engineered in the gRNA to enable recruitment of transcription activation domains linked to an MCP domain, such as in the bacterial dCas9-MS2/MCP-SoxS system (Dong *et al*, 2018) and the eukaryotic synergistic activation mediator (SAM) system (Konermann *et al*, 2015). We therefore tested CasTA-AsiA where the gRNA contains a MS2 domain in different stem-loops and where AsiA is tethered to MCP (i.e., dCas9-MS2/MCP-AsiA). While the MS2 hairpins did not affect the gRNA performance, we did not find that the SAM implementation of AsiA could activate gene activation (Appendix Fig S3). These results are in agreement with a previous observation that dCas9-

MS2/MCP-AsiA system was not a functional activator (Dong *et al*, 2018). We also did not find that a G32A mutant (DNA-binding disruption) (Griffith & Wolf, 2002) of the previously described SoxS activator in the dCas9-MS2/MCP-SoxS system (Dong *et al*, 2018) to be functional as a direct dCas9 fusion (i.e., dCas9-SoxS$_{G32A}$) (Fig 1C), potentially due to the instability of G32A mutant (Shah & Wolf, 2006). These results highlight potential mechanistic and performance differences between CRISPRa systems where the activation domain is directly fused to dCas9 versus tethered via the MS2-MCP system.

## Directed evolution and characterization of the dCas9-AsiA transcriptional activator

To increase the dynamic range and performance of dCas9-AsiA-mediated transcriptional activation, we performed a series of directed evolution studies using our screening–selection platform. A dCas9-AsiA variant library was constructed by error-prone PCR of AsiA, with each AsiA variant having on average two randomly distributed residue mutations (Appendix Fig S4). We screened

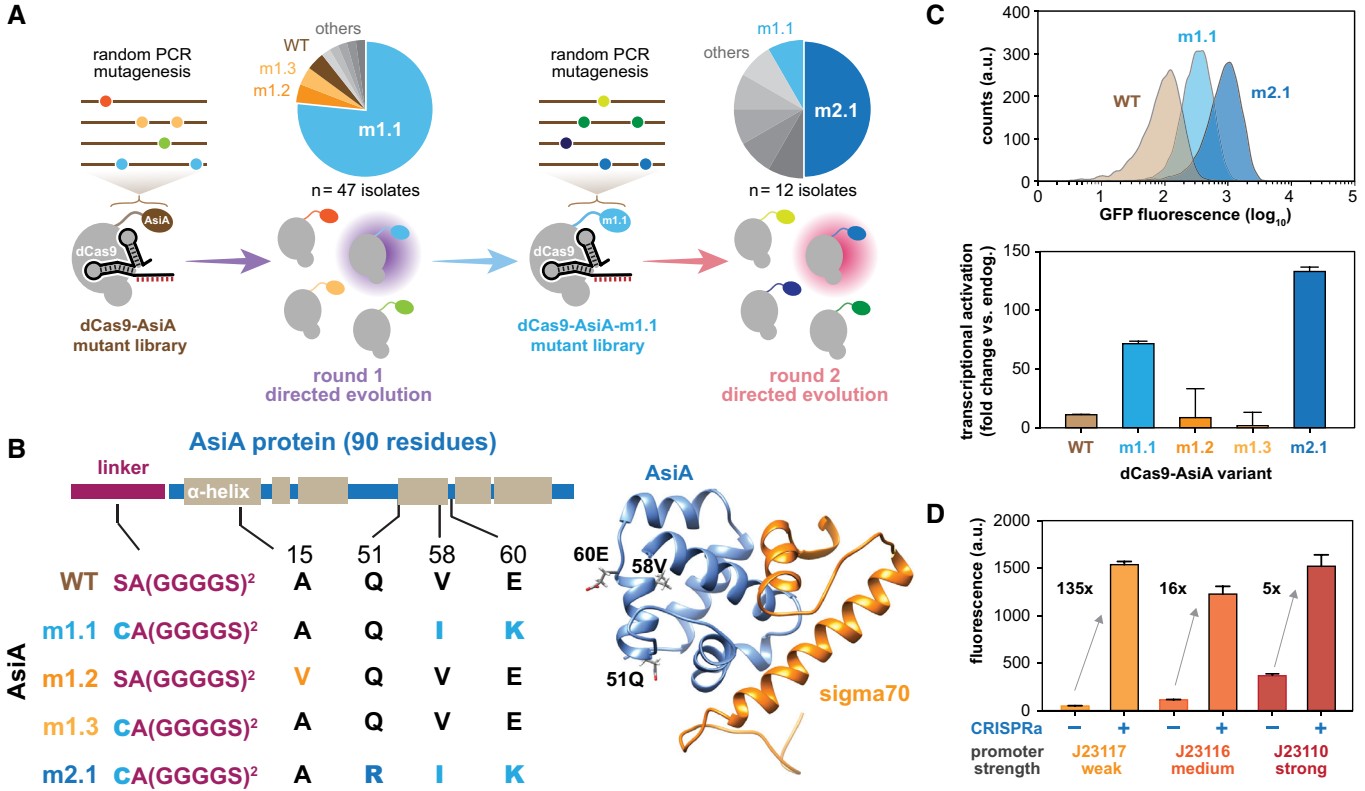

**Figure 2. Directed evolution of dCas9-AsiA for higher potency.**

A   Schematics of two rounds of directed evolution to improve potency of dCas9-AsiA. Pie charts show frequencies of dCas9-AsiA variants identified from each round.

B   Mutations found in enriched AsiA variants and their positions along the AsiA secondary structure (left). Crystal structure of wild-type AsiA (blue) interfaced with region 4 of σ70 (orange) are shown (right).

C   Distribution of fluorescence signal of the GFP reporter induced by different dCas9-AsiA variants (top). Fold induction by different dCas9-AsiA variants is shown (bottom).

D   CRISPRa induction of promoters with varying basal expression levels. CRISPRa-, basal expression of the promoter; CRISPRa+, expression activated by dCas9-AsiA-m2.1 and associated gRNA-H4.

Data information: Data shown are 3 biological replicates with ± SEM.

~5 × 10^8 AsiA mutant variants for improved transcriptional activation on antibiotic selection plates (Fig 2A, Appendix Fig S2). The resulting colonies were individually isolated, and plasmids encoding the dCas9-AsiA variants were extracted and transformed into cells expressing a gRNA and GFP reporter for re-validation (Appendix Table S4). Of 47 colonies isolated and characterized, one variant (m1.1) was found most enriched (> 75% of the time), while several other variants (m1.2, m1.3) were also identified at lower frequency (Fig 2A and B). The most abundant variant m1.1 after selection also mediated the highest GFP activation (Fig 2C). The m1.1 variant contained two key mutations on AsiA (V58I, E60K). An additional mutation (S1C) on the peptide linker was also found, which likely arose during the cloning steps of the directed evolution protocol. Interestingly, the AsiA mutations occurred within the middle of the anti-σ factor protein and are structurally away from the interface that binds to σ70 (Fig 2B) (Lambert *et al*, 2004). AsiA binds to sigma factors through the first helix structure (residues 1–20) (Minakhin *et al*, 2001). Hence, mutations in m1.1 may not be directly involved in binding to sigma factors but possibly induced structural change of AsiA, leading to higher activation. This m1.1

variant significantly increased the fold in transcriptional activation to ~70 compared to ~10 fold using the wild-type AsiA (Fig 2C). We then performed another round of directed evolution on m1.1 and screened for additional mutants with further improvements (Fig 2A). From 10^7 variants, validation and characterization of the resulting colonies revealed an additional mutant (m2.1) to be significantly enriched in the population with >135-fold activation (Fig 2B and C). The m2.1 variant contained an additional Q51R mutation, which also faced away from σ70 similar to the other m1.1 mutations.

We next explored the activation potential of dCas9-AsiA-m2.1 (CasTA2.1) for targeting promoters with different basal expression levels and at different CasTA2.1 expression levels. We observed that transcriptional activation across weak to strong promoters reached similar saturating levels and at the same optimal gRNA targeting distance (Fig 2D, Appendix Fig S5A and B). Accordingly, the fold induction inversely correlated with the basal promoter strength. To investigate the rules for gRNA designs at finer resolution, we constructed gRNAs targeting all NGG positions in the weak promoter (BBa_J23117) except for ones overlapping with σ70

binding sites and paired them with CasTA2.1 to mediate gene activation. We found an additional peak of activation at around 100 bps to TSS (Appendix Fig S5C). Similar periodicity of optimal gRNA targeting was recently observed in the dCas9-MS2/MCP-SoxS system (Fontana *et al*, 2020). However, CasTA2.1 has a generally broader activation window. For gRNAs that we tested with distances of more than 100 bp from the TSS, all leaded to gene activation from 10- to 288-fold. These 10 gRNAs targeted promoter regions across more than 150 bps, suggesting a flexible window for effective gRNA designs. Transcriptional or translational enhancement of the expression of CasTA1.0 or 2.1 could also increase activation of the target gene (Appendix Fig S5D), thus providing different options to tune the overall system.

Since AsiA binds and sequesters the host σ70, overexpression of AsiA may become toxic to the cell (Minakhin & Severinov, 2005). We therefore quantified the toxicity of dCas9-AsiA in our system. Overexpression of CasTA1.0 or 2.1 under aTc induction did not have significant impact on cellular growth rate beyond the basal fitness burden of dCas9 overexpression alone (Appendix Fig S6). Doubling times during exponential growth were generally unaffected under CasTA overexpression, while stationary cell density was somewhat impacted. To gain a higher resolution of the effects of CasTA on the endogenous transcriptome, we performed RNAseq on cells with CasTA1.0 and CasTA2.1, relative to ancestral control cells (Appendix Fig S7). We found that CasTA2.1 mediated higher gene activation on the GFP target without loss of specificity genome-wide compared to cells with CasTA1.0 (Appendix Fig S7A) or ancestral cells (Appendix Fig S7B). Upon overexpression of CasTA2.1, we observed upregulation of some low-expression endogenous genes (Appendix Fig S7C). These off-target gene activations may be the result of non-specific dCas9 binding to other genomic loci, which has been reported previously (Cho *et al*, 2018; Cui *et al*, 2018). This hypothesis is supported by the fact that strong off-targets (fold change > 30) were regulated by not just σ70 but also other σ factors, which would not be influenced by AsiA alone (Appendix Fig S7C). Notably, the fold induction of the GFP targets was also higher under CasTA2.1 overexpression (Appendix Fig S5D), which highlights a trade-off between higher target activation and increased off-targets in this CRISPRa system.

## Utility of dCas9-AsiA for multi-gene and library-scale transcriptional regulation

To explore whether CasTA can be used to regulate endogenous genomic targets, we first inserted a GFP reporter into the genome and showed that CasTA2.1 can upregulate the expression of this chromosomal reporter (Fig 3A). We then selected 10 endogenous genes and demonstrated that five of them could be highly upregulated (from 5 to 200-fold) using CasTA2.1 (Fig 3B, Appendix Fig S8, Appendix Table S6). Only one gRNA was designed for each gene using a search window of 190 ± 20 bp from the TSS in our attempt (Appendix Table S5), suggesting the ease of deploying CasTA2.1 toward chromosomal targets. However, we anticipate that optimization of gRNA designs may be necessary for different genomic targets (Bikard *et al*, 2013; Dong *et al*, 2018). We further explored whether CasTA2.1 can be used as a transcriptional repressor and for simultaneous CRISPRa/CRISPRi transcriptional modulation. We found that gRNAs (gRNA-H7 to gRNA-H10) positioned near the TSS or within the gene body of the target GFP reporter could efficiently inhibit gene expression using the CasTA2.1 protein, including both strands of the target DNA (Fig 3C). When two different gRNAs were designed to target two reporter genes for concurrent activation and repression, we observed simultaneous CRISPRa and CRISPRi using CasTA2.1 at efficiencies similar to applying CRISPRa or CRISPRi separately (Fig 3D), which highlights its potential utility for multiplexed gene modulation of regulatory networks in a single cell.

Development of complex synthetic genetic circuits requires diverse regulatory parts with tunable dynamic range (Slusarczyk *et al*, 2012). However, the number of inducible promoters with defined expression range is currently limited for many applications in synthetic biology. We previously developed a promoter library from metagenomic sequences with varying species-specific constitutive expression levels (Johns *et al*, 2018; Dataset EV1). We therefore explored whether such a constitutive promoter library could be turned into an inducible promoter library using our CRISPRa system (Fig 4A). We designed two gRNAs spaced ~150 bp apart targeting the constant regulatory region upstream of the variable regulatory sequences of each promoter and screened for subsets of promoters that could be upregulated by CasTA2.1. The expression level from all promoters in the library with and without CasTA2.1 was quantified by targeted RNAseq (to obtain RNA transcript for each promoter) and DNAseq (to normalize for plasmid copy numbers across the library) as previously described (Yim *et al*, 2019) (Appendix Fig S9A, Materials and Methods; Dataset EV2). Of ~8,000 promoters characterized, we identified thousands of promoters that were activated by CasTA2.1 with at least one of the gRNAs (Fig 4B, Appendix Fig S9B). Among them, several hundred had a high level of induction (> 10-fold) across 2 orders of magnitude in basal expression level (Fig 4C). In general, more promoters were activated with the distal gRNA (gRNA-H23), although interestingly the proximal gRNA (gRNA-H22) also resulted in CRISPRi activity in some promoters (Appendix Fig S9B). The phylogenetic origin and sequence composition of these inducible promoters were diverse, which will facilitate their use for assembly of large genetic circuits with minimal recurrent sequence motifs (Appendix Fig S9C). This library of CasTA-inducible promoters greatly expands the repertoire of regulatory parts that can be activated with one or two gRNAs by CRISPRa for more complex genetic circuits in various synthetic biology applications.

## Portability of dCas9-AsiA to other bacteria species

Since homologs of the T4 AsiA protein are widely found in many different phages that infect diverse bacteria (Fig 5A), we hypothesized that our dCas9-AsiA system could be ported to other bacteria with greater possibility of success and minimal re-optimization. We chose two bacterial species *Salmonella enterica* and *Klebsiella oxytoca* of clinical and bioindustrial significance (Kao *et al*, 2003; Coburn *et al*, 2007) to test our CasTA system. Each of the three plasmids (CasTA, gRNA, and reporter) was transformed into the two species. We first confirmed that dCas9 was functional in these two species by using a gRNA targeting for repression of a reporter GFP gene (i.e., CRISPRi) activity (Fig 5B). We then tested CRISPRa using the dCas9-AsiA wild type and CasTA 2.1 systems with the appropriate gRNA and GFP reporter. We were surprised to find that CasTA2.1 showed significant GFP activation in both species, but dCas9-AsiA wild type did not (Fig 5C). It is interesting to note that

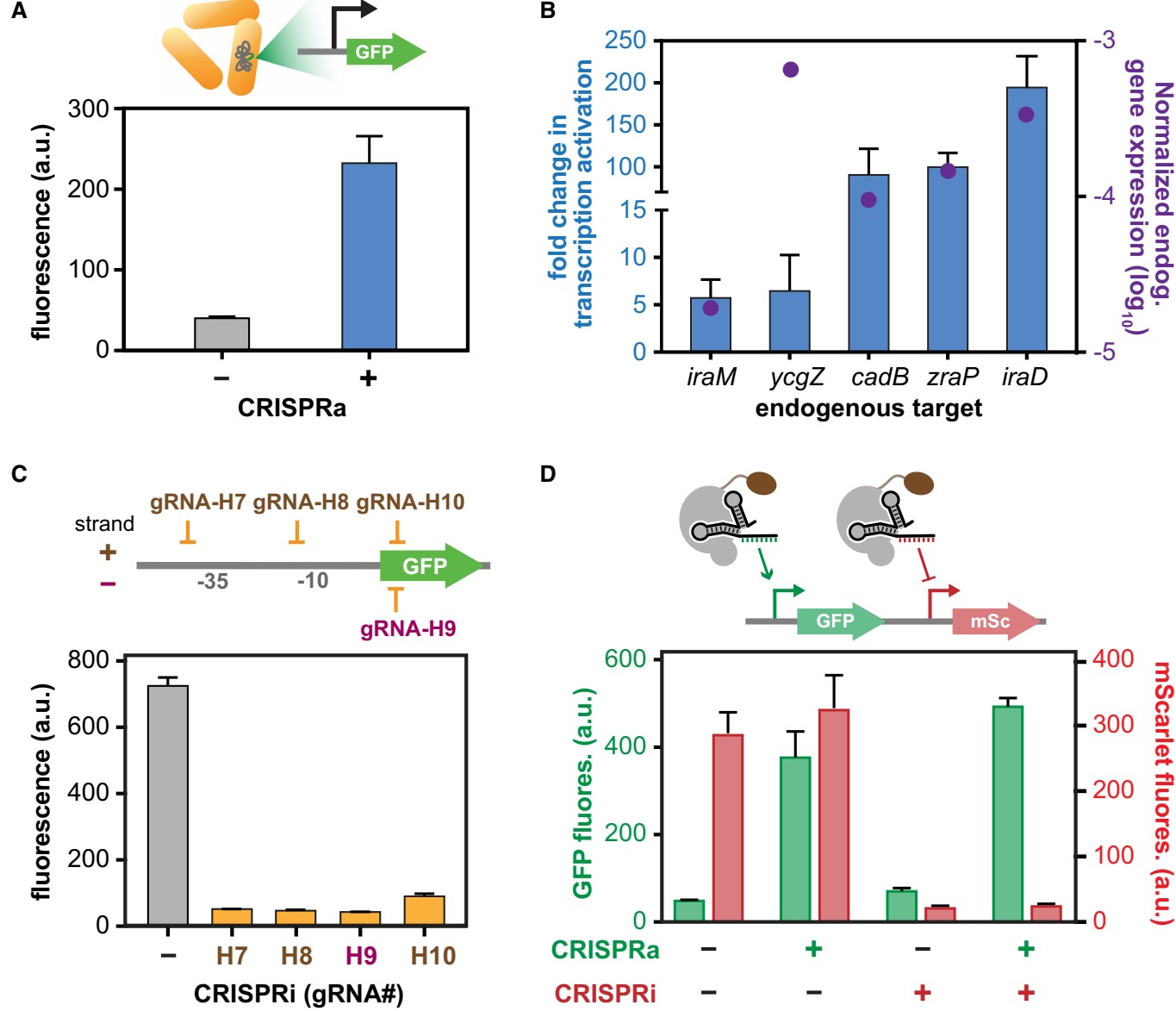

**Figure 3. Evolved CasTA2.1 could activate genomic targets and mediate multiplexed gene activation and repression.**

A   CasTA2.1 upregulates a genomically inserted GFP reporter.

B   Chromosomal gene targets activated by CasTA2.1 with bars showing the activation fold change and dots showing basal expression of each gene. Expression was quantified using RT-qPCR.

C   CasTA2.1 can mediate CRISPRi with appropriate gRNA designs by positioning different gRNAs relative to the target gene. A non-specific gRNA is used as the negative control.

D   Demonstration of multiplexed CRISPRa and CRISPRi using CasTA2.1 on a reporter containing GFP and mScarlet. Parental cells had low basal GFP and high basal mScarlet expression.

Data information: Data shown are 3–4 biological replicates with ± SEM.

AsiA from *Salmonella* phage SG1 shares the same residues at positions 50–61 as the *E. coli* T4 phage, while the *Klebsiella* phage F48 had some differences especially at residues 51–53, 57, and 59, which all face away from the binding surface to σ70. Notably, residues 51–53 and 57–61 of AsiA appear to be more variable across phylogenetically diverse phages (Fig 5A), which are also the key residue regions mutated in m2.1 (Q51R, V58I, E60K) from our directed

evolution experiments. In fact, some of the mutant residues in CasTA2.1 are also found in natural AsiA variants, suggesting that the mutations that we identified might mediate conserved molecular interactions leading to improved gene activation. Together, these results demonstrate that the CasTA system can be ported into other bacteria, although additional mutations may be required through directed evolution or testing of natural homologs.

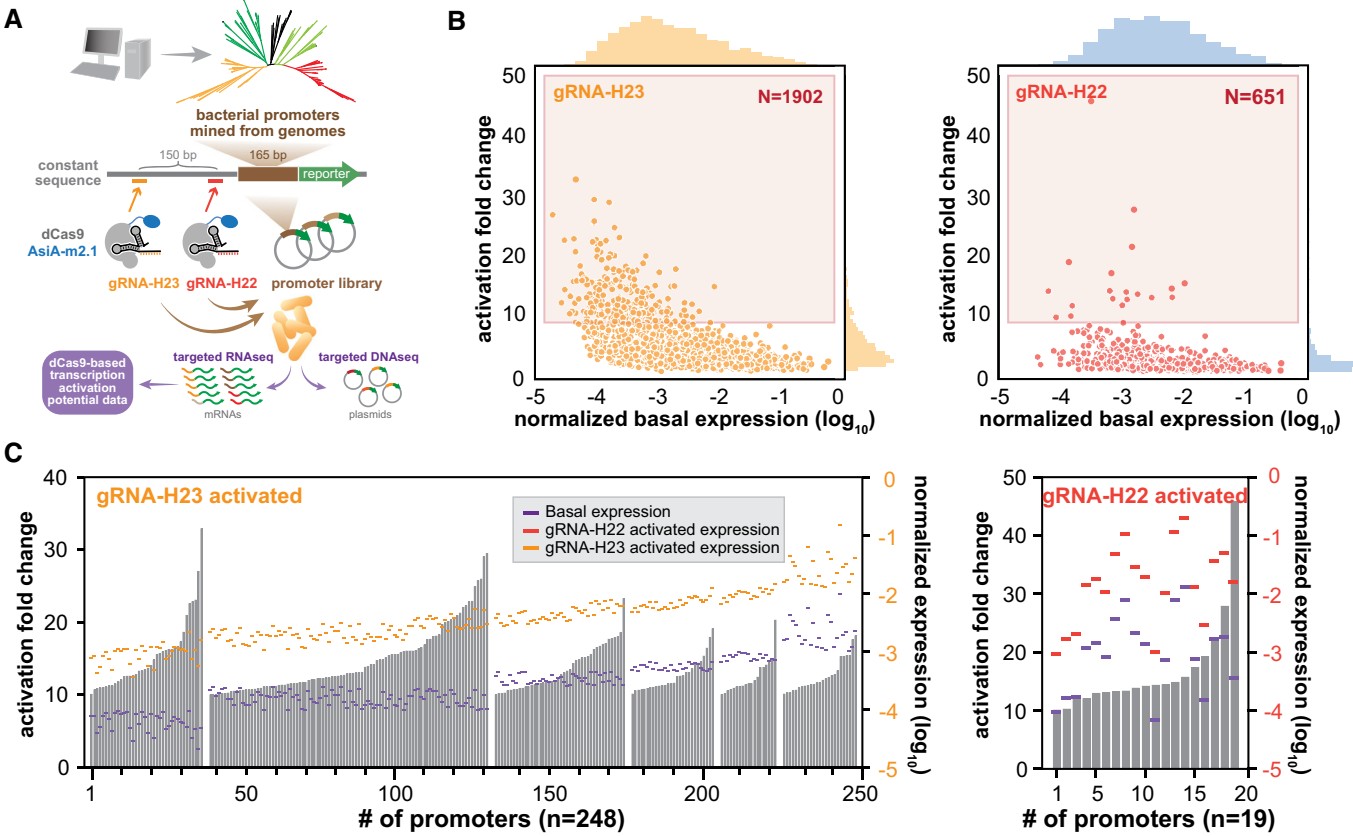

**Figure 4. Multiplex reporter assay to identify inducible promoters using CasTA2.1.**

A  Schematics of construction and screening platform to characterize a library of CRISPRa-mediated inducible promoters by targeted RNAseq and DNAseq.

B  Promoters significantly activated by CasTA2.1 using gRNA-H23 (left) or gRNA-H22 (right) are plotted with basal expression level on *x*-axis against fold activation by CRISPRa on *y*-axis. *N* is the total number of promoters shown. Red box corresponds to highly activated promoters (fold change > 10).

C  Highly activated promoters (> 10-fold) using gRNA-H23 (left) or gRNA-H22 (right) are plotted. Basal expression levels (purple lines), induced expression level (orange lines on left or red lines on right, activated with gRNA-H23 or gRNA-H22, respectively) and induced fold changes (gray bars) are shown. Promoters were first ranked and grouped by basal expression levels. Within each group, promoters were ranked by activation fold changes.

# Discussion

CRISPRa is a powerful approach to elucidate the cellular and genetic basis of various biological phenomena in bacteria and higher organisms (Konermann *et al*, 2015; Liu *et al*, 2018). While numerous CRISPRa systems with high efficiency have been established in eukaryotes, reliable bacterial CRISPRa tools have been slow to develop. The screening–selection platform to identify active CRISPR-Cas-mediated transcriptional activation established in this work provides a highly modular and portable system to engineer new bacterial CRISPRa systems for enhancing potency of dCas9 activators, modifying gRNA designs, or screening for targetable promoters. We demonstrated here that CasTA2.1 could activate endogenous genes and enable genome-scale gain-of-function studies in bacteria, which have not been shown in prior studies. Given that not all genomic targets are activated with the same potency, we anticipate that some targets may require additional gRNA optimizations to improve activation.

The capability to multiplex CRISPRa and CRISPRi on multiple targets can facilitate combinatorial screens, which have been challenging to implement in traditional genetic screens in bacteria with transposon mutagenesis libraries or metagenomic expression libraries (Hu & Coates, 2005; Yaung *et al*, 2015). The inducible promoter library characterized in this study constitutes a useful catalog of promoters with varying basal expression level and defined fold activation upon CRISPRa induction. This library could serve as a community resource for promoter optimization in metabolic engineering applications or construction of complex genetic circuits. Importantly, multiple promoters could be induced by the same gRNA, which provide opportunities to synchronize modulation of multiple network nodes at the same time.

Sequencing of environmental metagenomic libraries has found many biosynthetic pathways encoding biomolecules with various biotherapeutic potential (Donia *et al*, 2014; Cohen *et al*, 2017). However, the majority of these biosynthetic gene clusters remain silent when transformed into common heterologous expression hosts such as *E. coli* (Iqbal *et al*, 2016). As such, CRISPR activation tools could be useful to reawaken these silent gene clusters. Our results indicate that CasTA2.1 is compatible with promoters from diverse bacterial origins, suggesting its potential utility in

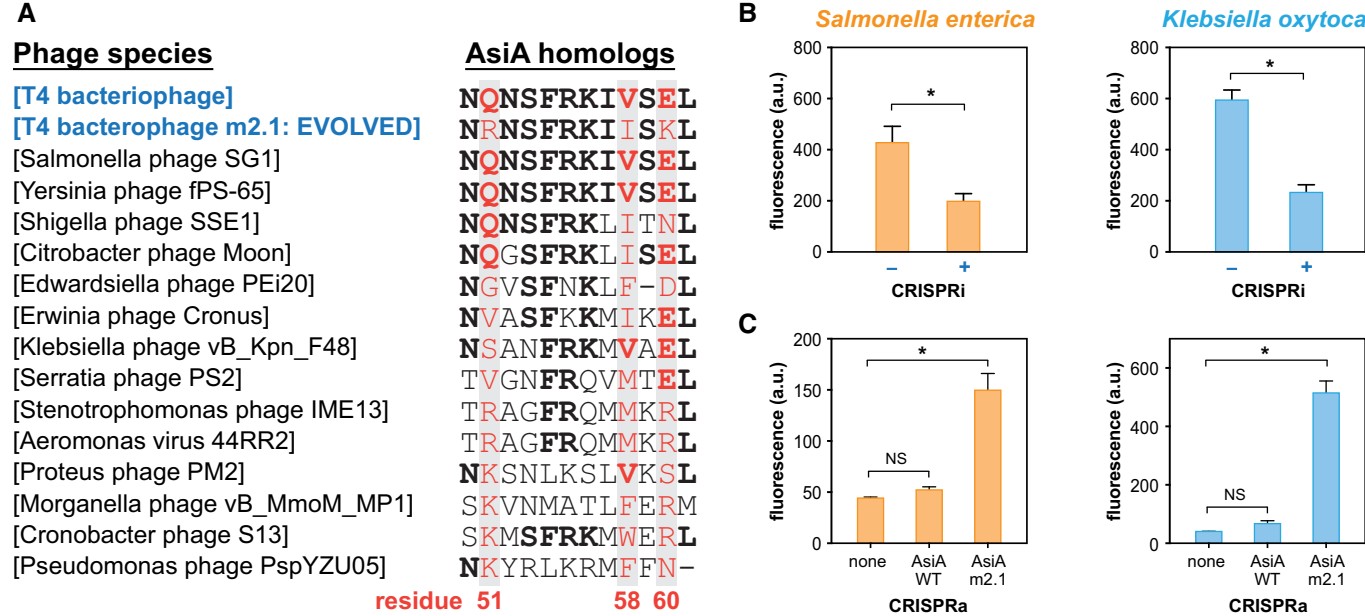

**Figure 5. Evolved CasTA functions in multiple bacterial species.**

A   Multiple sequence alignment of AsiA homologs from different phage genomes at residue positions 50–61. Highlighted red residues indicate positions that are mutated in AsiA-m2.1.

B   CRISPRi in *S. enterica* and *K. oxytoca* using CasTA2.1.

C   CRISPRa in *S. enterica* and *K. oxytoca* using CasTA1.0 (dCas9-AsiA_wt), CasTA2.1 (dCas9-AsiA_m2.1) or ancestral strain with basal promoter expression (none).

Data information: All data are 3–4 biological replicates with ± SEM. *Student's *t*-test, *P* < 0.0001 NS = non-significant.

activating cryptic gene clusters. In addition, increasing the portability of bacterial CRISPRa systems can greatly improve our ability to manipulate non-model organisms. In combination with *in situ* microbiome engineering tools (Brophy *et al*, 2018; Ronda *et al*, 2019), we envision that CasTA and similar technologies

(Peters *et al*, 2019) could modulate gene expression in a precise and programmable fashion across diverse bacterial communities to elucidate fundamental processes underlying microbial ecology and provide useful applications in the emerging field of microbiome engineering.

# Materials and Methods

**Reagents and Tools table**

| Reagent/Resource | Reference or source | Identifier or catalog number |
|---|---|---|
| **Experimental models** | | |
| BW25113 (*E. coli*) | CGSC | 7636 |
| JEN202 (*E. coli*) | Luciano lab (Bikard *et al*, 2013 PMID: 23761437) | |
| Serovar Typhi Ty2 (*S. enterica*) | ATCC | 700931 |
| M5A1 (*K. oxytoca*) | ATCC | 7342 |
| **Recombinant DNA** | | |
| pdCas9-bacteria | Addgene | 44249 |
| pgRNA-bacteria | Addgene | 44251 |
| pWJ89 | Luciano lab (Bikard *et al*, 2013 PMID: 23761437) | |
| pWJ96 | Luciano lab (Bikard *et al*, 2013 PMID: 23761437) | |
| pWJ97 | Luciano lab (Bikard *et al*, 2013 PMID: 23761437) | |
| pEB2-mScarlet-I | Addgene | 104007 |

**Reagent and Tools table**  (continued)

| Reagent/Resource | Reference or source | Identifier or catalog number |
|---|---|---|
| RS7003 promoter library | Johns *et al*, 2018 PMID: 30052624 | |
| pOSIP-Kan | St-Pierre *et al*, 2013 PMID: 24050148 | |
| pdCas9-linker | This study | |
| pdCas9-AsiA | This study | |
| pdCas9-AsiA_m1.1 | This study | |
| pdCas9-AsiA_m2.1 | This study | |
| pHH39 | This study | |
| Additional plasmids and more information | This study | Appendix Table S2 |
| **Oligonucleotides and other sequence-based reagents** | | |
| guide RNAs_N20 | This study | Appendix Table S5 |
| Q_RT_PCR primers | This study | Appendix Table S6 |
| **Chemicals, enzymes and other reagents** | | |
| Q5® High-Fidelity 2X Master Mix | New England Biolabs | M0492S |
| NEBuilder® HiFi DNA Assembly Master Mix | New England BioLabs | E2621 |
| T4 DNA ligase | New England BioLabs | M0202 |
| T4 Polynucleotide kinase | New England BioLabs | M0201 |
| SuperScript III Reverse Transcriptase | Invitrogen | 18080-093 |
| KAPA SYBR FAST qPCR Master Mix | Kapa Biosystems | KK4602 |
| Maxima reverse transcriptase | Thermo Scientific | EP0742 |
| **Software** | | |
| Geneious v11.1 | https://www.geneious.com/ | |
| Benchling | https://www.benchling.com/ | |
| Python 3.6.0 | https://www.python.org/ | |
| Bowtie 2 | (Langmead & Salzberg, 2012 PMID:22388286) https://github.com/BenLangmead/bowtie2 | |
| HTseq | (Anders *et al*, 2015 PMID:25260700) https://htseq.readthedocs.io/en/master/ | |
| BBmerge | (Bushnell *et al*, 2017 PMID:29073143) https://jgi.doe.gov/data-and-tools/bbtools/bb-tools-user-guide/bbmerge-guide/ | |
| DRAFTS | (Yim *et al*, 2019 PMID:31464371) https://github.com/ssyim/DRAFTS | |
| **Other** | | |
| GeneMorph II EZClone Domain Mutagenesis Kit | Agilent Technologies | 200552 |
| RNA Clean & Concentrator Kits | Zymo Research | R1030 |
| DNA Clean & Concentrator Kits | Zymo Research | D4013 |
| Ribo-Zero rRNA Removal Kit (Bacteria) | Illumina | |
| NEBNext Ultra Directional RNA Library Prep Kit | New England BioLabs | E7760 |
| Illumina NextSeq 500/550 mid output kit v2/v2.5 (150/300 cycles) | Illumina | Cat #20024904/ 20024905 |
| PrepGem bacteria kit | MicroGEM | PBA0100 |
| BD FACS Aria II | BD Biosciences | |
| Guava® InCyte | MilliporeSigma | |
| Synergy H1 plate reader | BioTek | |
| CFX96 Touch Real-Time PCR machine | Bio-Rad | |

## Methods and Protocols

### Strains and culturing conditions

*E. coli* strains and other bacterial species used in the study are listed in Appendix Table S1, and all *E. coli* strains are derived from the MG1655 parental background. Cells were grown in rich LB medium at 37°C with agitation unless stated otherwise. For plasmid transformation, general protocols were followed, and plasmids were maintained under antibiotics selection at all times. For constructing genomic insertions, the GFP expression cassette amplified from pWJ89 (Bikard *et al*, 2013) was cloned between multiple cloning sites of pOSIP-Kan and inserted chromosomally following the clonetegration method (St-Pierre *et al*, 2013). For the antibiotic selection and induction of target genes, the following concentrations were used: carbenicillin (Carb) 50 μg/ml, chloramphenicol (Cam) 20 μg/ml, kanamycin (Kan) 50 μg/ml, spectinomycin (Spec) 50 μg/ml, Bleocin (Bleo) 5 μg/ml, and anhydrotetracycline (aTc) 100 ng/ml. For induction of target genes, aTc was added to the culture at the exponential growth phase for 4 h before cells were harvested for characterization.

### Construction of plasmids

The dCas9 fusion library was constructed based on the pdCas9-bacteria plasmid (Addgene #44249). Linker sequences (SAGGGGSGGGGS) and fusion candidates were either amplified from DNA synthesized *de novo* (IDT gBlocks®) or *E. coli* genomic DNA and subcloned after the dCas9 sequence in the pdCad9-bacteria plasmid (Addgene #44249). All guide RNA plasmids (pgRNA-H1 to pgRNA-H21) were constructed from the pgRNA-bacteria plasmid (Addgene #44251), using inverted PCR and blunt-end ligation to modify the N20 seed sequences. For dual gRNA plasmids (pgRNA-H4H5, pgRNA-H4H11), each gRNA was built separately and jointed subsequently. GFP reporter plasmids (pWJ89, pWJ96, pWJ97) were gifts from the Marraffini lab at Rockefeller University (Bikard *et al*, 2013). The promoter region upstream of the GFP reporter in pWJ89 was amplified for constructing other antibiotic reporter plasmids (pHH34-37). The GFP-mScarlet reporter plasmid (pHH39) was constructed by cloning the mScarlet gene from pEB2-mScarlet-I (Addgene #104007) under the WJ97 promoter and joined with the weak GFP expression cassette from pWJ89. For screening the inducible metagenomic promoter library (RS7003) (Johns *et al*, 2018), gRNA-H22 and gRNA-H23 expression cassettes were joined with dCas9-AsiA_m2.1 separately, resulting pHH40 and pHH41. Cloning was done by Gibson assembly if not otherwise noted in all cases. Plasmids used and associated details are listed in Appendix Table S2.

### Development of CasTA screening platform

The dCas9 fusion library, gRNAs, and reporter genes were built on 3 different compatible plasmids (dCas9: p15A, Cam resistance; gRNA: ColE1, Carb resistance; reporter: SC101, Kan resistance), so they can be transformed and propagated within the same cell (Appendix Fig S1). To use a antibiotic resistance gene as a reporter, we tested different antibiotic genes and modulated degradation rate (fusion with ssrA tag: AANDENYALAA) for selective stringency (Appendix Fig S2A and B). Dual selective reporters (Kan and Bleo) were constructed, which decrease the escape rate by 10 fold (Appendix Fig S2C and D).

### Directed evolution of dCas9-AsiA using CasTA screening platform

We mutagenized the wild-type AsiA region of dCas9-AsiA using the GeneMorph II EZClone Domain Mutagenesis Kit (Agilent Technologies), following the manufacture's protocol.

- In brief, 50 ng of parental template DNA was used for amplification with error-prone DNA polymerase (Mutazyme II). Under this condition, the AsiA region contains on average ~ 2 nucleotides changes per variant after PCR mutagenesis (Appendix Fig S4).
- In the first round of directed evolution, the dCas9-AsiA mutant library was transformed to the cells expressing gRNA-H4 and dual selective reporters (pHH37 and pHH38). ~ $5 \times 10^8$ transformants were grown under 0.2× regular Kan concentration and 2× regular Bleo concentration.
- Grown colonies were harvested and propagated together with Cam selection to maintain solely the dCas9-AsiA variant plasmids.
- The dCas9-AsiA plasmids were subsequently extracted and transformed to cells containing pgRNA-H4 and pWJ89.
- Individual colonies were Sanger sequenced to identify the mutations in AsiA and characterized based on GFP intensity (Appendix Table S4). The background fluorescence was measured using the parental strain (BW25113), and auto-fluorescence was subtracted from the fluorescence readings of all samples. Fold change of fluorescence was normalized to cells expressing the GFP only plasmid (pWJ89).
- The dCas9-AsiA_m1.1 plasmid from the most abundant mutant variant was extracted and transformed to the GFP reporter strain (containing pgRNA-H4 and pWJ89) again to verify fluorescent intensity (Fig 2C).
- In the second round of directed evolution, the dCas9_AsiA_m1.1 variant was used as a template to generate additional variants following the same conditions.
- The second generation of the dCas9-AsiA mutant library was transformed to GFP reporter cells, containing pgRNA-H4 and pWJ89 as described above.
- We enriched the top 0.1% of highest GFP expression from the population of $1 \times 10^7$ transformants using fluorescence activated cell sorting (BD FACS Aria II).
- Post-sorted cell population was plated on selective LB again to obtain clonal colonies, and individual colonies were picked for Sanger sequencing and measurement of GFP intensity.

### Quantification of gene expression induced by CasTA

To examine CRISPRa on genomic targets, pdCas9-AsiA_m2.1 was transformed along with gRNA constructs (gRNA-H12 to gRNA-H21, Appendix Table S5) designed for each gene (Appendix Table S6). Cells expressing dCas9-AsiA_m2.1 and a non-specific gRNA (gRNA-H4) were used as controls.

- After CRISPRa induction with 100 ng/ml aTc, cells were harvested for RNA extraction following the RNAsnap protocol (Stead *et al*, 2012).
- After column purification (RNA Clean & Concentrator Kits, Zymo Research), total RNA was reverse-transcribed into cDNA using random hexmers (SuperScript III Reverse Transcriptase, Invitrogen).
- Quantitative PCR was performed on each sample with gene-specific primers (Appendix Table S6) using the KAPA SYBR FAST qPCR Master Mix (Kapa Biosystems). The *rrsA* gene was selected as the housekeeping gene to normalize expression between samples.

For whole-transcriptome analysis of CRISPRa specificity,

- we extracted total RNA from the samples as described above and depleted rRNAs using Ribo-Zero rRNA removal-Bacteria kit (Illumina).
- RNA libraries were prepared using the NEBNext Ultra Directional RNA Library Prep Kit (New England BioLabs) and sequenced on the Illumina NextSeq platform (Mid-Output Kit, 150 cycles).
- The raw reads were aligned to the reference genome (BW25113) using Bowtie 2 (Langmead & Salzberg, 2012), and the read counts of each gene were quantified by HTseq (Anders *et al*, 2015). Expression level of individual genes was normalized by total read counts within each sample.

### Screening for CRISPRa-mediated inducible promoters

The Metagenomic promoter library (RS7003) was derived from Johns *et al* (2018).

- About 8,000 regulatory elements were transformed to cells expressing dCas9-AsiA_m2.1 and either gRNA-H22, gRNA-H23, or genomic targeting gRNA-H24 (Appendix Table S5).
- After CRISPRa induction, four biological replicates were harvested to measure promoter activity. A constitutive promoter without CRISPRa induction (Appendix Table S7) was spiked in the cell populations for normalizing expression levels between samples.
- Total RNA was extracted and purified as previously described. Gene-specific primers were used for cDNA generation (Maxima reverse transcriptase, Thermo Scientific), and RNA sequencing library was prepared by ligation with the common adaptor primer for downstream sequencing (Yim *et al*, 2019).
- To quantify abundance of each promoter in the library, plasmid DNA from each sample was also extracted using PrepGem bacteria kit (MicroGEM) and used to generate a DNA amplicon sequencing library.
- Both RNA and DNA libraries were sequenced on the Illumina NextSeq platform (Mid-output kit, 300 cycles).
- Sequencing reads from DNA and RNA libraries were merged by BBmerge and filtered out low-quality reads (Bushnell *et al*, 2017).
- Custom pipeline that was previously described (Yim *et al*, 2019) was adopted to identify sequencing reads corresponding to each promoter. Expression level of each promoter was quantified by determining the ratio of RNA abundance over DNA abundance. To compare across samples, expression levels were normalized to the same spiked-in control promoter in each sample. Fold change in CRISPRa induced gene expression was calculated by dividing by the reporter expression of control cells containing dCas9-AsiA_m2.1 and a genomic targeting gRNA-H24.

# Data availability

The sequencing data associated with this study are stored and available at NCBI SRA under PRJNA637809 (https://www.ncbi.nlm.nih.gov/bioproject/PRJNA637809/).

**Expanded View** for this article is available online.

## Acknowledgements

We thank members of the Wang laboratory for comments and discussions on the manuscript and experiments, especially S. S. Yim and N. Johns for advice and discussions on the inducible promoter library experiments. H.H.W. acknowledges specific funding support from NSF (MCB-1453219), NIH (1U01GM110714, 1DP5OD009172), DoD ONR (N00014-15-1-2704), and the Sloan Foundation (FR2015-65795) for this work.

## Author contributions

H-IH and HHW developed the initial concept; H-IH, JRF, and JC. performed experiments and analyzed the results; and HHW supervised the work throughout. H-IH and HHW wrote the manuscript with inputs from all authors.

## Conflict of interest

The authors declare that they have no conflict of interest.

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
