## [Review Process File · Molecular Systems Biology]

Programmable CRISPR-Cas transcriptional activation in bacteria

Hsing-I Ho, Jennifer Fang, Jacky Cheung, and Harris Wang

DOI: [10.15252/msb.20199427](https://doi.org/10.15252/msb.20199427)

Corresponding author(s): Harris Wang (hw2429@columbia.edu) , Harris Wang (hw2429@columbia.edu)

Review Timeline:	Submission Date:	19th Dec 19
	Editorial Decision:	7th Feb 20
	Revision Received:	7th Jun 20
	Accepted:	16th Jun 20

Editor: Maria Polychronidou

Transaction Report:

Manuscript Number: MSB-19-9427

Title: Programmable and portable CRISPR-Cas transcriptional activation in bacteria

Thank you again for submitting your work to Molecular Systems Biology. We have now heard back from the three referees who agreed to evaluate your study. Overall, the reviewers are quite supportive. They raise however a series of concerns, which we would ask you to address in a revision. The recommendations of the reviewers are rather clear and I think there is no need to repeat any of the points listed below.

REFeree REPORTS

Reviewer #1:

In this manuscript Ho et al. engineer a novel transcriptional activator for bacteria by fusing dCas9 to the phage T4 AsiA activator. While dCas9 activators have been previously engineered in bacteria, they remain limited to small sets of promoters, narrow windows of binding positions and weak activation. The authors screened many possible activators and possible designs to converge on a protein fusion of dCas9-AsiA that they further evolved to improve its activity. The authors perform a detailed characterization of the properties of their CasTA, including its ability to activate various promoters, its toxicity and off-target activity. In particular the author show that many promoters from a library of microbial promoters can be activated using guides targeting the same constant upstream region, providing a possibly useful set of activable promoters for synthetic biology. The CasTA was finally shown to be active in two closely related bacterial species Salmonella and Klebsiella. The manuscript is well written, the experiments well performed and the data supports the conclusions. The contribution of this manuscript is an important step in making CRISPRa a viable technology for microbiologists.

Major points

The authors argue that the activation window of CasTA relative to the promoter is different from previous systems which is indeed supported by the data in Fig 1E and FigS5. However the characterization of this window is very limited (only 4 points over a window of ~200nt). It would be very valuable to provide a thorough characterization of the activation provided by CasTA2.1 at various distances and orientation from the promoter in a much more dense fashion. This would greatly improve the usability of the tool by helping researchers identify the best candidate target positions in front of their promoter of interest. How easy is it to find a good target position? If the window is too narrow, many promoters might not have a PAM in the right position. The authors were able to obtain weak to strong activation of 5 endogenous targets. How were the genes and targets chosen? How many targets were tested for each gene? What was the orientation and distance to the promoter for the ones that worked?

The different activation window also suggests a different mechanism of activation compared to

previous dCas9 activators. Can the authors provide more information about the activation mechanism of AsiA and comment on this phenomenon?

Reviewer #2:

CRISPRa is a method to activate gene expression in a targeted and inducible fashion, by recruiting dCas9-fused transcription factors (TF) to the promoters of the gene of interest using a gRNA. The authors have developed a screening method to identify bacterial TF suitable for CRISPRa in *E. coli*. Using their platform, they found phage protein AsiA to work well, optimized its activation capacity and used it to create a modular, inter-species portable bacterial CRISPRa system, expanding the CRISPRa method to the bacterial domain. The 'portable' claim is a bit far fetched as only 2 organisms similar to *E. coli* that allow for replication of the already made plasmids were tested.

General remarks

Good work, opening up reprogrammable gene transcription manipulation techniques for bacteria in general. Enjoyed the paper.

Major points

- Figure 1C: of what are fold changes depicted? Normalized fluorescence? qPCR RNA abundance? Please add this information to the Figure legend.
 - Figure 1E: Figure 1A suggests the two gRNAs with highest TA fold change (H4 and H5) target a different strand than the other gRNAs. Did the authors consider the possibility that the high TA fold change might simply be due to the strand that was targeted, rather than the position relative to the TSS? It would be good to test this systematically, both distance from TSS and strand as this is quite important for CRISPRi (Qi et al. Cell 2013).
 - The authors suggest that the optimized dCas9-AsiA variants m1.1 and m2.1 do not thank their improved activation capabilities to changed interaction with the sigma-70 factor, as their mutations occur away from the binding interface. Can the authors speculate on how this might work instead? How come these variants have higher activation fold changes?
 - Fig. 2D, at which distance to the TSS was the sgRNA targeted in these assays?
 - Figure S7: Are there no replicates? If there are, please indicate in figure legend and also what is shown (average over replicates?).
 - Fig. 3B, how is fold change in transcription determined? RT-qPCR?
 - Figure 3C: a previous study (Qi et al. (2013) Cell) showed that for CRISPRi, when targeting the gene body, targeting the non-template strand is substantially more efficient than the template strand. This seems to contrast the result in Figure 3C, with similar efficiencies for the two strands. Can the authors speculate on this difference?
 - Figure S8B: Can the authors offer an explanation as to why gRNA-H22 can result in CRISPRi activity for some genes, opposed to CRISPRa activity for others (as was meant to happen)?
 - Figure 4
 - o Do I understand correctly all transcripts are of equal length? If not, should the authors not correct for this bias, e.g. using the Transcripts Per kilobase Million (TPM)?
 - o 4B: Do the authors have an explanation as to why there appears to be an inverse correlation between normalized basal expression and activation fold change, at least for gRNA-H23?
 - o 4C: How are the promoters on the x-axis ordered and why? Please explain in the legend.
- Fig. S8 suggests that that the described CRISPRa system might work in bacilli. If the authors could show that the systems works or not works in the model Gram positive organism *Bacillus subtilis* for instance, it would really increase the impact of the study. Without this, I would suggest to change

the title and remove 'portable'.

Minor points

- Line numbers missing in manuscript
- Introduction: 2002 & 2013 do not seem to be 'recent' ("With the recent discovery of new DNA-binding proteins")
- Introduction: authors speak of eukaryotic CRISPRi, but used references do not describe them.
- How did the authors come to their "list of transcriptional activator candidates"?
- Figure S7
 - o Consider transforming reads using a variance-stabilizing transformation (vst) or on rlog scale (DESeq2 package in R). This decouples variance from the mean, thereby getting rid of the increased (biased) spread for lower read counts.
 - o You can get rid of the p-values, as these merely indicate the slope is significantly different from 0, which is not the point to be made.
 - o Indicate in Figure legend what the "r" in the plots stands for (Pearson correlation coefficient?)
- Introduction
 - o "A potential path for cellular engineering is therefore" (instead of "therefore is")
 - o "Zinc-finger TFs" instead of "Zinc-fingers TF"
 - o Introduction: "programmable and targeted transcriptional repression (CRISPRi) or activation (CRISPRa)" → remove "e.g." 2x
 - o Missing space: "exists(Chavez et al, 2016)."
 - o Space too many (before period): "bacterial CRISPRa have been shown ."
 - o "holoenzyme and regulate gene expression" (instead of "regulating")
 - o "an MS2 domain", "an MCP-fused transcription factor", "an S. pyogenes dCas9", "an MCP domain" (instead of "a")
- Results
 - o "the gRNA was expressed constitutively" (instead of "constitutive")
 - o "defined expression range" (instead of "ranged")
- Figure legends
 - o 1C: "AisA" should be "AsiA"
 - o 2D: "basal" should be "basal"
- Discussion
 - o "Sequencing of environmental metagenomic libraries has" (instead of "have")

Reviewer #3:

Summary

This manuscript reports the development of a new CRISPRa system for programmable gene activation in bacteria. The authors fused the phage anti-sigma factor AsiA to dCas9 and found that it could activate reporter gene expression. They optimized the activator by directed evolution, demonstrated that it could be used for multiplexed gene activation and repression, screened for activity against a promoter library, and demonstrated activity in multiple bacterial species. There are only a few implementations of CRISPRa in bacteria and there is substantial value in developing alternative systems that can be tested in different applications. This manuscript will have a significant impact on the transcriptional control and bacterial engineering fields. I recommend that the manuscript be published after the authors consider the minor comments below.

Major Comments

No major comments.

Minor comments

1. On page 3, the authors write: "Furthermore, most of these prior studies have only demonstrated CRISPRa in laboratory *E. coli* strains and activity in other bacteria is unknown." In addition to Liu 2019 (cited by the authors), there are a few reports of CRISPRa in other bacteria that could be cited here, although it is true that there is vast room for improvement in generalizable, portable tools. See DOIs: [10.1021/acssynbio.7b00293](https://doi.org/10.1021/acssynbio.7b00293), [10.1186/s12934-018-0867-1](https://doi.org/10.1186/s12934-018-0867-1), [10.1093/nar/gkz072](https://doi.org/10.1093/nar/gkz072)
2. Is the complete sequence of the promoter for the reporter gene in Fig 1A available? Is one of the pWJ plasmids listed in Table S3 the reporter in Fig 1A? Are these pWJ plasmids the same as those reported in Bikard 2013? More detail is needed in the figure legends and/or methods to provide clear answers to these questions. Table S3 should include a column indicating whether the plasmids were generated for this study or obtained from another source and include a reference.
3. On page 4, the authors write: "Among the transcription activation modules screened, we found a phage protein, AsiA, that upregulated the reporter gene expression to a level comparable to the previously identified dCas9- ω activator (Bikard et al, 2013), although at a different optimal spacing distance (Fig. 1C-D)." As the authors note, the dCas9- ω activator requires a Δ rpz strain background. Presumably a Δ rpz strain was used for this experiment, but it is not clearly described in the figure legend, main text, or methods. There is a Δ rpz strain listed in Table S2. Was the same Δ rpz strain used for all activators screened in Fig 1C or just for dCas9- ω ? I don't think it is necessary that the comparisons between activators shown in Fig 1C all occurred in the same strain, but it should be clear exactly what was done here.
4. On page 5, the authors write: "Transcriptional activation by dCas9-AsiA (dubbed CasTA1.0) is seen across a wide window along the target regulatory region, reaching up to 12-fold at ~200 base pairs (bp) from the TSS (Fig. 1E)." The authors could use more precise language here. Liu 2019 (cited by the authors) noted significant changes in activation over relatively short-range distance changes with a periodicity of 10 bases. Here the authors are testing gRNA sites spaced out over >>10 bases, and they should acknowledge that they do not know if there are similar periodicity effects with dCas9-AsiA.
5. On page 5, the authors write: "When directly fused to dCas9 with a peptide linker, AsiA upregulated gene expression of a GFP reporter, with the magnitude of activation tunable via design of the gRNA." What exactly do the authors mean by "design" here? Do they mean varying the target sequence position?
6. On page 5, the authors write: "We also did not find that a G32A mutant (DNA binding disruption) (Griffith & Wolf, 2002) of the previously described SoxS activator in the dCas9-MS2/MCP-SoxS system (Dong et al, 2018) to be functional as a direct dCas9 fusion (i.e. dCas9-SoxSG32A) (Fig. 1C)." Why was G32A SoxS chosen here? This mutant was not among the mutant SoxS variants tested for CRISPRa by Dong 2018. G32A is one of several SoxS mutants reported by Griffith and Wolf to have defective DNA binding. However, the DNA binding defects observed in Griffith and Wolf could be due to disrupting the DNA binding interface or globally destabilizing the protein. Is it possible that SoxS G32A is simply misfolded? There is data in Shah and Wolf 2006 consistent with this possibility ([doi:10.1111/j.1365-2958.2006.05086.x](https://doi.org/10.1111/j.1365-2958.2006.05086.x)). I do not think additional experiments with

dCas9-SoxS mutants are necessary, but some explanation would be useful.

7. Related to the point above, the candidate activators from Fig 1 are listed in Table S1. This table should include a column with a reference for each of the candidate activator proteins.

8. In Figure S3B, the authors show data for dCas9-AsiA with various MS2 gRNA designs and MS2-AsiA. It is somewhat surprising that dCas9-AsiA/MS2-AsiA/gRNA-MS2 shows such weak activity compared to dCas9-AsiA/gRNA-MS2. Can the authors offer any interpretation? Is expression of MS2-AsiA problematic?

9. In Figure S5, there is an apparent discrepancy between the fold-activation values in panels A-B and the values in panel C. The fold activation values in panel S5C seem unusually low (~2-3X) compared to the values in S5A-B (ranging up to ~150X). What RBS was used for the promoter variants in S5A-B? What minimal promoter was used with the RBS variants in S5C?

10. On page 7, the authors write: "These off-target gene activations may be the result of non-specific dCas9 binding to other genomic loci, which has been reported previously (Cho et al, 2018; Cui et al, 2018). This hypothesis is supported by the fact that strong off-targets (fold change >30) were regulated by not just $\sigma 70$ but also other σ factors (Fig. S7C)." The second sentence is confusing. Can the authors elaborate? Why is it significant that some of the off-targets are regulated by other σ factors? Is the logic that if AsiA was interacting with off-target genes independently of dCas9, the expectation would be that it would only affect $\sigma 70$ promoters?

11. In Figure S7, the figure legend for B and C requires more explanation. What exactly is the difference between B and C? The authors indicate that in panel C dCas9-AsiA was overexpressed. Can the authors specify which promoters were used for "expression" (panel B) and "overexpression" (panel C)?

12. On page 7, the authors write: "We then demonstrated that five genes in the genome could be upregulated (by up to 200-fold) using CasTA2.1 (Fig. 3B, Table S6). One gRNA was designed for each gene using a search window of 200{plus minus}20bp from the TSS." Is this statement an accurate reflection of the success rate for endogenous gene targets? Did the authors attempt to activate more than 5 endogenous genes? How often were the authors able to activate an endogenous gene target on the first try with a single gRNA in the specified search window? How were these particular 5 endogenous gene targets chosen?

Reviewer #1:

In this manuscript Ho et al. engineer a novel transcriptional activator for bacteria by fusing dCas9 to the phage T4 AsiA activator. While dCas9 activators have been previously engineered in bacteria, they remain limited to small sets of promoters, narrow windows of binding positions and weak activation. The authors screened many possible activators and possible designs to converge on a protein fusion of dCas9-AsiA that they further evolved to improve its activity. The authors perform a detailed characterization of the properties of their CasTA, including its ability to activate various promoters, its toxicity and off-target activity. In particular the author show that many promoters from a library of microbial promoters can be activated using guides targeting the same constant upstream region, providing a possibly useful set of activable promoters for synthetic biology. The CasTA was finally shown to be active in two closely related bacterial species Salmonella and Klebsiella. The manuscript is well written, the experiments well performed and the data supports the conclusions. The contribution of this manuscript is an important step in making CRISPRa a viable technology for microbiologists.

We thank the reviewer for their enthusiasm for this work and have addressed the concerns in detail in the response below and in edits to the revised manuscript.

Major points

R1.1. The authors argue that the activation window of CasTA relative to the promoter is different from previous systems which is indeed supported by the data in Fig 1E and FigS5. However the characterization of this window is very limited (only 4 points over a window of ~200nt). It would be very valuable to provide a thorough characterization of the activation provided by CasTA2.1 at various distances and orientation from the promoter in a much more dense fashion. This would greatly improve the usability of the tool by helping researchers identify the best candidate target positions in front of their promoter of interest. How easy is it to find a good target position? If the window is too narrow, many promoters might not have a PAM in the right position.

A1.1. In order to more thoroughly characterize the activation window of our CasTA system, we constructed another 14 gRNA designs targeting all the NGG sites in both orientations across the entire promoter region and paired with CasTA2.1. In short, the activation window of CasTA2.1 still shows distinct features from previous systems in more distal optimal targeting position (> 100 bps from TSS versus 80 bps) and a wider effective window expanding more than 100 bps, suggesting a more flexible CasTA system for designing gRNAs. These results are discussed in main text (Line202-211) and presented in a new subpanel Appendix Fig S5C.

R1.2. The authors were able to obtain weak to strong activation of 5 endogenous targets. How were the genes and targets chosen? How many targets were tested for

each gene? What was the orientation and distance to the promoter for the ones that worked?

A1.2. We selected 10 genes in total that were of interest for metabolic engineering applications and with varying endogenous expression levels and NGG sites within a search window of 190 ± 20 bp from the TSS. We included all results in the additional Appendix Fig S8 and Main text (Line 237-241). Importantly, only “1” gRNA was designed and examined for activating each genomic target. As such, these results should provide insights on how relatively easy for regulating genomic targets through our system without any prior knowledge or optimization. Nonetheless, we agree that predicting and generalize the rule for targeting any endogenous genes based on the genomic context would be interesting in future investigations.

R1.3. The different activation window also suggests a different mechanism of activation compared to previous dCas9 activators. Can the authors provide more information about the activation mechanism of AsiA and comment on this phenomenon?

A1.3. We elaborated more on the activation mechanism of AsiA in the Main text (Line 147-151). We speculated that dCas9-AsiA differs from other CasTAs because AsiA is a phage anti-sigma 70 proteins whereas other activators are *E. coli* endogenous transcription factors. As such, the AsiA protein has evolved to bind sigma 70 strongly and outcompeting endogenous transcriptional machinery. The strong binding affinity may contribute to high local enrichment of sigma 70s, resulting in its wider activation window.

Reviewer #2:

*CRISPRa is a method to activate gene expression in a targeted and inducible fashion, by recruiting dCas9-fused transcription factors (TF) to the promoters of the gene of interest using a gRNA. The authors have developed a screening method to identify bacterial TF suitable for CRISPRa in *E. coli*. Using their platform, they found phage protein AsiA to work well, optimized its activation capacity and used it to create a modular, inter-species portable bacterial CRISPRa system, expanding the CRISPRa method to the bacterial domain. The 'portable' claim is a bit far fetched as only 2 organisms similar to *E. coli* that allow for replication of the already made plasmids were tested.*

General remarks

Good work, opening up reprogrammable gene transcription manipulation techniques for bacteria in general. Enjoyed the paper.

We thank the reviewer for their enthusiasm for this work and have addressed the concerns in detail in the response below and in edits to the revised manuscript.

Major points

R2.1. Figure 1C: of what are fold changes depicted? Normalized fluorescence? qPCR RNA abundance? Please add this information to the Figure legend.

A2.1. The fold changes shown in Fig. 1C were normalized fluorescence fold change compared to control cells. We have provided this information in the legend.

R2.2. Figure 1E: Figure 1A suggests the two gRNAs with highest TA fold change (H4 and H5) target a different strand than the other gRNAs. Did the authors consider the possibility that the high TA fold change might simply be due to the strand that was targeted, rather than the position relative to the TSS? It would be good to test this systematically, both distance from TSS and strand as this is quite important for CRISPRi (Qi et al. Cell 2013).

A2.2. We do not believe the fold activation is determined by the strand orientation based on the results from other bacterial CasTA systems (Bikard et al. 2013, Dong et al. 2018) In addition, we did not find evidence supporting such hypothesis in experiments responding to R1.1 and in experiments when we targeted endogenous genes.

R2.3. The authors suggest that the optimized dCas9-AsiA variants m1.1 and m2.1 do not thank their improved activation capabilities to changed interaction with the sigma-70 factor, as their mutations occur away from the binding interface. Can the authors speculate on how this might work instead? How come these variants have higher activation fold changes?

A2.3. We hypothesized that these residues were not directly involved in binding to sigma 70 since they do not locate in the binding surface. However, these mutations may contribute to overall conformation change of AsiA, resulting in altered interaction with sigma 70 and higher fold activation. We included these discussions in the Main text (Line 189-190).

R2.4. Fig. 2D, at which distance to the TSS was the sgRNA targeted in these assays?

A2.4. We used gRNA H4, which targeted around 190 bps from TSS in all these assays. This point has been clarified in the figure legend.

R2.5. Figure S7: Are there no replicates? If there are, please indicate in figure legend and also what is shown (average over replicates?).

A2.5. Each data point from Figure S7 is a single biological sample from the RNA-seq experiment. This point has been clarified in the figure legend.

R2.6. Fig. 3B, how is fold change in transcription determined? RT-qPCR?

A2.6. Fold change in transcription was quantified by RT-qPCR, and we now specify this in the legend.

R2.7. Figure 3C: a previous study (Qi et al. (2013) Cell) showed that for CRISPRi, when targeting the gene body, targeting the non-template strand is substantially more efficient than the template strand. This seems to contrast the result in Figure 3C, with similar efficiencies for the two strands. Can the authors speculate on this difference?

A2.7. In our CRISPRi results, targeting the non-template strand is still significantly more effective than the template strand, consistent with studies from other group. We believe that some gRNAs are capable of inhibiting gene expression even targeting the template strand, which has been shown previously (Bikard et al., 2013).

R2.8. Figure S8B: Can the authors offer an explanation as to why gRNA-H22 can result in CRISPRi activity for some genes, opposed to CRISPRa activity for others (as was meant to happen)?

A2.8. H22 targets the region closer to promoter libraries, so it may interfere transcription for genes with more distal transcription start sites in the library. As we showed in Appendix Fig S5C, gRNAs target positions with distance <100 bps from the TSS all led to inhibitory effect on gene expression.

R2.9.1. Figure 4

Do I understand correctly all transcripts are of equal length? If not, should the authors not correct for this bias, e.g. using the Transcripts Per kilobase Million (TPM)?

A2.9.1. All the transcripts were first reverse transcribed by a universal primer aligned to the beginning of the downstream GFP. As a result, the cDNA library has a highest size limit of 180 bps, and the whole length of transcript was covered by paired-end NGS sequencing of 300 cycles. Therefore, each merged read count corresponded to single transcript.

R2.9.2. 4B: Do the authors have an explanation as to why there appears to be an inverse correlation between normalized basal expression and activation fold change, at least for gRNA-H23?

A2.9.2. As shown in multiple CRISPRa systems, the fold activation usually inversely correlates with endogenous expression level, also consistent with our results in Fig. 2D and Fig. 4. Highly expressed genes may have limited room for further enhancement due to already saturated transcription machinery and limited resources for transcription in cells (e.g. the amount of RNAPs).

R2.9.3. 4C: How are the promoters on the x-axis ordered and why? Please explain in the legend.

A2.9.3. Promoters were first filtered and grouped by their basal expression levels. Within each group, promoters were then ranked by activation fold changes. Details are now included in the legend.

R2.10 Fig. S8 suggests that that the described CRISPRa system might work in bacilli. If the authors could show that the systems works or not works in the model Gram positive organism Bacillus subtilis for instance, it would really increase the impact of the study. Without this, I would suggest to change the title and remove 'portable'.

A2.10. In light of the reviewer's suggestion, we have removed "portable" from the title.

Minor points

- Line numbers missing in manuscript

We have included line numbers in the revised manuscript.

- Introduction: 2002 & 2013 do not seem to be 'recent' ("With the recent discovery of new DNA-binding proteins")

We have removed the word 'recent' and 'now' for potential confusions.

- Introduction: authors speak of eukaryotic CRISPRi, but used references do not describe them.

The eukaryotic CRISPRi was also demonstrated in mammalian cells in Qi et al 2013. We modified the position of relevant citations to exclude confusions.

- How did the authors come to their "list of transcriptional activator candidates"?

We chose these candidates for reported binding to RNAPs. Relevant literatures were summarized in the Appendix Table S3.

- Figure S7

o Consider transforming reads using a variance-stabilizing transformation (vst) or on rlog scale (DESeq2 package in R). This decouples variance from the mean, thereby getting rid of the increased (biased) spread for lower read counts.

We did not transform the reads as they were from single experiments.

o You can get rid of the p-values, as these merely indicate the slope is significantly different from 0, which is not the point to be made.

We have modified the figure as suggested.

o Indicate in Figure legend what the "r" in the plots stands for (Pearson correlation coefficient?)

It is Pearson's correlation here. We have now indicated this information in the legend.

- Introduction

- o "A potential path for cellular engineering is therefore" (instead of "therefore is")
- o "Zinc-finger TFs" instead of "Zinc-fingers TF"
- o Introduction: "programmable and targeted transcriptional repression (CRISPRi) or activation (CRISPRa)" ∅ remove "e.g." 2x
- o Missing space: "exists(Chavez et al, 2016)."
- o Space too many (before period): "bacterial CRISPRa have been shown ."
- o "holoenzyme and regulate gene expression" (instead of "regulating")
- o "an MS2 domain", "an MCP-fused transcription factor", "an *S. pyogenes* dCas9", "an MCP domain" (instead of "a")

- Results

- o "the gRNA was expressed constitutively" (instead of "constitutive")
- o "defined expression range" (instead of "ranged")

- Figure legends

- o 1C: "AisA" should be "AsiA"
- o 2D: "basal" should be "basal"

- Discussion

- o "Sequencing of environmental metagenomic libraries has" (instead of "have")

We thank the reviewer for their suggestions and modified the texts in the revised manuscript.

Reviewer #3:

Summary

This manuscript reports the development of a new CRISPRa system for programmable gene activation in bacteria. The authors fused the phage anti-sigma factor AsiA to dCas9 and found that it could activate reporter gene expression. They optimized the activator by directed evolution, demonstrated that it could be used for multiplexed gene activation and repression, screened for activity against a promoter library, and demonstrated activity in multiple bacterial species. There are only a few implementations of CRISPRa in bacteria and there is substantial value in developing alternative systems that can be tested in different applications. This manuscript will have a significant impact on the transcriptional control and bacterial engineering fields. I recommend that the manuscript be published after the authors consider the minor comments below.

We thank the reviewer for their enthusiasm for this work and have addressed the concerns in detail in the response below and in edits to the revised manuscript.

Major Comments

No major comments.

Minor comments

R3.1. On page 3, the authors write: "Furthermore, most of these prior studies have only demonstrated CRISPRa in laboratory *E. coli* strains and activity in other bacteria is unknown." In addition to Liu 2019 (cited by the authors), there are a few reports of CRISPRa in other bacteria that could be cited here, although it is true that there is vast room for improvement in generalizable, portable tools. See DOIs: 10.1021/acssynbio.7b00293, 10.1186/s12934-018-0867-1, 10.1093/nar/gkz072

A3.1. We thank the reviewer for the suggestions and included these references as suggested.

R3.2. Is the complete sequence of the promoter for the reporter gene in Fig 1A available? Is one of the pWJ plasmids listed in Table S3 the reporter in Fig 1A? Are these pWJ plasmids the same as those reported in Bikard 2013? More detail is needed in the figure legends and/or methods to provide clear answers to these questions. Table S3 should include a column indicating whether the plasmids were generated for this study or obtained from another source and include a reference.

A3.2. The pWJ plasmids used in the paper were from Bikard et al 2013 without modification on the promoter sequences. We have provided the information about the source in the Reagents and Tools Table as suggested.

R3.3. On page 4, the authors write: "Among the transcription activation modules screened, we found a phage protein, AsiA, that upregulated the reporter gene expression to a level comparable to the previously identified dCas9- ω activator (Bikard et al, 2013), although at a different optimal spacing distance (Fig. 1C-D)." As the authors note, the dCas9- ω activator requires a Δ rpoZ strain background. Presumably a Δ rpoZ strain was used for this experiment, but it is not clearly described in the figure legend, main text, or methods. There is a Δ rpoZ strain listed in Table S2. Was the same Δ rpoZ strain used for all activators screened in Fig 1C or just for dCas9- ω ? I don't think it is necessary that the comparisons between activators shown in Fig 1C all occurred in the same strain, but it should be clear exactly what was done here.

A3.3 The dCas9- ω was characterized in a Δ rpoZ strain background, and dCas9-AsiA was characterized in a wild-type background (BW25113). We have provided details about used strains in the legend as suggested.

R3.4. On page 5, the authors write: "Transcriptional activation by dCas9-AsiA (dubbed CasTA1.0) is seen across a wide window along the target regulatory region, reaching up to 12-fold at ~200 base pairs (bp) from the TSS (Fig. 1E)." The authors could use more precise language here. Liu 2019 (cited by the authors) noted significant changes in activation over relatively short-range distance changes with a periodicity of 10 bases. Here the authors are testing gRNA sites spaced out over \gg 10 bases, and they should acknowledge that they do not know if there are similar periodicity effects with dCas9-

AsiA.

A3.4. As described in A1.1., we characterized the system more thoroughly with more gRNAs. We did observe similar periodic effects on peaks of gene activation, but with more pervasive activation window that is unique in this system.

R3.5. On page 5, the authors write: "When directly fused to dCas9 with a peptide linker, AsiA upregulated gene expression of a GFP reporter, with the magnitude of activation tunable via design of the gRNA." What exactly do the authors mean by "design" here? Do they mean varying the target sequence position?

A3.5. We apologize for the confusion. We meant designing gRNAs with different targeting positions. We modified the text to "via design of the gRNA targeting positions" for clarity (Line 142-143).

R3.6. On page 5, the authors write: "We also did not find that a G32A mutant (DNA binding disruption) (Griffith & Wolf, 2002) of the previously described SoxS activator in the dCas9-MS2/MCP-SoxS system (Dong et al, 2018) to be functional as a direct dCas9 fusion (i.e. dCas9-SoxSG32A) (Fig. 1C)." Why was G32A SoxS chosen here? This mutant was not among the mutant SoxS variants tested for CRISPRa by Dong 2018. G32A is one of several SoxS mutants reported by Griffith and Wolf to have defective DNA binding. However, the DNA binding defects observed in Griffith and Wolf could be due to disrupting the DNA binding interface or globally destabilizing the protein. Is it possible that SoxS G32A is simply misfolded? There is data in Shah and Wolf 2006 consistent with this possibility (doi:10.1111/j.1365-2958.2006.05086.x). I do not think additional experiments with dCas9-SoxS mutants are necessary, but some explanation would be useful.

A3.6. The rationale for picking SoxS G32A, a DNA binding defective mutant, as our candidate is to minimize non-specific binding to other genomic loci. It is possible that the instability of this mutant results in the failure of CRISPRa. We added text and relevant literature (Line 167-168) to better explain this possible explanation as suggested by the reviewer.

R3.7. Related to the point above, the candidate activators from Fig 1 are listed in Table S1. This table should include a column with a reference for each of the candidate activator proteins.

A3.7. We have included the references in Appendix Table S3 as suggested.

R3.8. In Figure S3B, the authors show data for dCas9-AsiA with various MS2 gRNA designs and MS2-AsiA. It is somewhat surprising that dCas9-AsiA/MS2-AsiA/gRNA-MS2 shows such weak activity compared to dCas9-AsiA/gRNA-MS2. Can the authors offer any interpretation? Is expression of MS2-AsiA problematic?

A3.8. The weak activation of dCas9-AsiA/MS2-AsiA/gRNA-MS2 may be due to the toxic effect from overexpressing MS2-AsiA proteins. Overwhelming cells with MS2-AsiA may have a dominant negative effect for sigma-70 sequestering. The design of dCas9/MS2-AsiA/gRNA-MS2 was found to be non-functional in the Dong et al, 2018 work, consistent with our results.

R3.9. In Figure S5, there is an apparent discrepancy between the fold-activation values in panels A-B and the values in panel C. The fold activation values in panel S5C seem unusually low (~2-3X) compared to the values in S5A-B (ranging up to ~150X). What RBS was used for the promoter variants in S5A-B? What minimal promoter was used with the RBS variants in S5C?

A3.9. The fold changes in panels A-B were compared with control cells (containing GFP plasmid only), whereas in panel S5D the comparison was between induction or not and different RBS strength. Due to leaky expression of dCas9-AsiA via Ptet regulation, GFP was somewhat up-regulated without induction as shown in Fig. S5D. We used strong RBS in S5A-B, and the same strong RBS sequence was used in S5D along with the weak RBS variant. Different RBS variants were all constructed under the same Ptet promoters.

R3.10. On page 7, the authors write: "These off-target gene activations may be the result of non-specific dCas9 binding to other genomic loci, which has been reported previously (Cho et al, 2018; Cui et al, 2018). This hypothesis is supported by the fact that strong off-targets (fold change >30) were regulated by not just $\sigma 70$ but also other σ factors (Fig. S7C)." The second sentence is confusing. Can the authors elaborate? Why is it significant that some of the off-targets are regulated by other σ factors? Is the logic that if AsiA was interacting with off-target genes independently of dCas9, the expectation would be that it would only affect $\sigma 70$ promoters?

A3.10. We hypothesized that the enrichment of dCas9-AsiA to other genomic loci facilitates sigma 70 to initiate transcription on non- sigma 70 promoters. Since AsiA primarily interacts with sigma 70, overexpression of AsiA alone should not affect non-sigma 70 promoters, suggesting the importance of non-specific binding of dCas9. We modified the second sentence to "This hypothesis is supported by the fact that strong off-targets (fold change >30) were regulated by not just $\sigma 70$ but also other σ factors, which would not be influenced by AsiA alone (Appendix Fig S7C)" (Line 227-229) for clarification.

R3.11. In Figure S7, the figure legend for B and C requires more explanation. What exactly is the difference between B and C? The authors indicate that in panel C dCas9-AsiA was overexpressed. Can the authors specify which promoters were used for "expression" (panel B) and "overexpression" (panel C)?

A3.11. As we showed in Appendix Fig S5D, CRISPRa was effective even under leaky expression of dCas9-AsiA. The "expression" in panel B refers to basal expression of

dCas9-AsiA (no ATC induction), and “overexpression” refers to dCas9-AsiA under ATC induction. We have clarified this information in the legend.

R3.12. On page 7, the authors write: "We then demonstrated that five genes in the genome could be upregulated (by up to 200-fold) using CasTA2.1 (Fig. 3B, Table S6). One gRNA was designed for each gene using a search window of 200{plus minus}20bp from the TSS." Is this statement an accurate reflection of the success rate for endogenous gene targets? Did the authors attempt to activate more than 5 endogenous genes? How often were the authors able to activate an endogenous gene target on the first try with a single gRNA in the specified search window? How were these particular 5 endogenous gene targets chosen?

We selected 10 genes in total that were of interest for metabolic engineering applications and with varying endogenous expression levels and NGG sites within a search window of 190 ± 20 bp from the TSS. We included all results in the additional Fig. S8 and Main text (Line 237-241). Importantly, only “1” gRNA was designed and examined for activating each genomic target. As such, these results should provide insights on the relative ease for regulating genomic targets through our system without any prior knowledge or optimization. Nonetheless, we agree that predicting and generalizing the rule for targeting any endogenous genes based on the genomic context would be interesting in future investigations.

Manuscript number: MSB-19-9427R, Programmable CRISPR-Cas transcriptional activation in bacteria

Thank you again for sending us your revised manuscript. We are now satisfied with the modifications made and I am pleased to inform you that your paper has been accepted for publication.

Corresponding Author Name: Harris Wang

Manuscript Number: MSB-19-9427R